# A Portable Biodevice to Monitor Salivary Conductivity for the Rapid Assessment of Fluid Status

**DOI:** 10.3390/jpm11060577

**Published:** 2021-06-19

**Authors:** Chun-Hao Chen, Yen-Pei Lu, An-Ting Lee, Chun-Wu Tung, Yuan-Hsiung Tsai, Hsin-Pei Tsay, Chih-Ting Lin, Jen-Tsung Yang

**Affiliations:** 1Department of Orthopedic Surgery, Chang Gung Memorial Hospital, Chiayi 61363, Taiwan; a0983109189@gmail.com; 2Taiwan Instrument Research Institute, National Applied Research Laboratories, Hsinchu 30261, Taiwan; ypl@narlabs.org.tw; 3Department of Anesthesiology, Chang Gung Memorial Hospital, Chiayi 61363, Taiwan; annielee@cgmh.org.tw; 4Department of Nephrology, Chang Gung Memorial Hospital, Chiayi 61363, Taiwan; P122219@cgmh.org.tw; 5Graduate Institute of Clinical Medical Sciences, Chang Gung University, Taoyuan 33302, Taiwan; 6Department of Diagnostic Radiology, Chang Gung Memorial Hospital, Chiayi 61363, Taiwan; russell.tsai@gmail.com; 7Department of Neurosurgery, Chang Gung Memorial Hospital, Chiayi 61363, Taiwan; thp191108@gmail.com; 8Graduate Institute of Electrical Engineering, National Taiwan University, Taipei 10617, Taiwan; 9College of Medicine, Chang Gung University, Taoyuan 33302, Taiwan

**Keywords:** salivary conductivity, fluid status, portable biodevice

## Abstract

The evaluation of fluid status can save adults from life-threatening conditions, but the current methods are invasive or time-consuming. Therefore, we developed a portable device for measuring salivary conductivity. This prospective observational study enrolled 20 volunteers with no history of systemic diseases. Participants were observed for 13 h, including water restriction for 12 h followed by rehydration with 1000 mL water within 1 h. Serum and urine biomarkers for fluid status, thirst scales, and salivary conductivity were collected during dehydration and rehydration. No significant differences in age, body mass index, glycohemoglobin, and estimated glomerular filtration rate were noted between sexes. Salivary conductivity increased after water restriction and decreased after rehydration. Similarly, urine osmolality, urine specific gravity, thirst intensity scales, and body weight followed the same trend and were statistically significant. The angiotensin-converting enzyme and aldosterone levels showed the same trend, without reaching statistical significance. The red blood cell count and hemoglobin concentration also followed the same trend. Analyzing the receiver operating characteristic curves, the area under the curve was 0.707 (95% confidence interval 0.542–0.873, *p* = 0.025). Using the Youden index, the optimal cutoff determined as 2678.09 μs/cm (sensitivity: 90%, specificity: 55%). This biodevice effectively screened dehydration among healthy adults.

## 1. Introduction

The regulation of fluid status is a complicated process that is meticulously conducted by the human body. The massive loss of body fluid can lead to an imbalance in fluid status and can sometimes progress to dehydration. Progressive negative changes in fluid status can directly affect many organ systems and cause life-threatening circulatory failure in severe clinical conditions [1]. Friedrich et al. [2] reported that dehydration is associated with a higher risk of disability and mortality among the elderly. It has also been described as a risk factor for falls in older adults and is considered a key factor associated with the performance of soccer athletes under stress [3,4]. Cognitive function and performance have been shown to improve after adequate water supplementation after dehydration [5,6]. Furthermore, several clinical studies have revealed that dehydration is highly related to mortality and morbidity [7,8,9]. However, early fluid imbalance may not result in obvious clinical symptoms, and an individual with dehydration may be asymptomatic before the onset of critical disease. Therefore, the early detection of fluid imbalance is important to prevent subsequent deterioration of the clinical condition.

The balance and disorders of water and electrolytes have central roles in the clinical conditions related to fluid status. Medical history and physical examination have been used for diagnosis of dehydration. However, both clinical history and physical examination cannot diagnose dehydration validly. Medical history can provide information of possible water depletion but cannot provide quantitative data objectively. Physical examination has been reported to not be valid in clinical practice due to its poor sensitivity in detection of dehydration [10,11,12]. Currently, several blood and urine biomarkers are being used for the evaluation of dehydration. Serum osmolality can indicate euvolemia based on the objective normal range of serum tonicity (275–295 mOsm/kg of water). Urine specific gravity and osmolality have also been reported as valid biomarkers for the evaluation of fluid status [13,14]. However, in clinical practice, continuous monitoring of serum osmolality is not feasible because of its invasiveness. Meanwhile, although the collection of urine specimens is simple and non-invasive, urine voiding is not available at any time. Additionally, extreme dehydration reduces the amount of urine, making the collection of urine more difficult. This limits the practicality of urine testing. Several clinical studies have used other non-invasive biomarkers, such as sweat and tears, for the evaluation of fluid status [15]. However, sweat collection is time-consuming and is limited to sweating participants. Conversely, the collection of tear specimens is affected by evaporation and differences in the collection methods [16].

In contrast, saliva can be collected directly using a simple method that does not require specific equipment and can be performed even in a non-clinical setting, making it a feasible option. Most importantly, saliva collection is a non-invasive procedure. Salivary conditions can reflect fluid status after fluid intake and stabilize within half a minute [17]. 

We recently developed a novel device for measuring saliva conductivity, which has been found to be associated with fluid status [18]. Data on saliva conductivity can be acquired immediately using a portable monitor system, which allows for a non-invasive and real-time evaluation of fluid status. This pilot study aimed to confirm the correlation between salivary conductivity and fluid status with acute dehydration and rehydration in healthy participants.

## 2. Materials and Methods

### 2.1. Ethical Statement

This longitudinal observational study complied with the guiding principles of the Declaration of Helsinki and was approved by the Medical Ethics Committee of Chang Gung Memorial Hospital (institutional review board number: 201700651B0). Before the study, all subjects signed an informed consent form. 

### 2.2. Subjects

Twenty healthy adults aged 18 years and above volunteered to participate in the study. They received regular annual health examinations at Chiayi Chung Gang Memorial Hospital. They were all non-smokers with no oral, dental, or systemic diseases during the study. Systemic disease concerned in this study any cardiovascular disease, peripheral vascular disease, renal disease, hepatic disease, infection episode and endocrine disease. They also did not take any prescription or over-the-counter medications during the study. The power of this study is 80% with a hypothetical effect size of 0.6. The required sample would be 20 participants. Post hoc computing of the power of study and effect size of all the outcomes was performed using the G-power statistical software.

### 2.3. Experimental Procedures

We explained the experimental design to all participants before the study, including saliva collection, hematological and biochemical examination, body composition analysis, and evaluation of thirst scale. Data collection was conducted at three continuous time points: at the beginning of the study, after 12 h of water restriction, and at 1 h after rehydration with 1000 mL of water. In this study, the participants were prohibited from drinking any fluid and eating food containing more than 500 mL fluid for 12 h to achieve a dehydration status. There was no limitation of their physical activities and the status of dehydration was confirmed by serum and urinary analysis after 12 h.

### 2.4. Saliva Collection and Analysis

Saliva specimens of all volunteers were collected orally at least 1 h after any intake. Saliva samples were assessed with a portable conductivity meter and a sensing probe, which were designed and fabricated as described in a previous study [18]. The miniaturized sensing electrode was placed in horizontally parallel and its sample-contact area reduced. Hence, the required amount of saliva was reduced to 50–500 μL for a test. The conductivity meter was pre-calibrated with the standard conductivity solution prior to the clinical study. Saliva samples were applied to the well of the sensing probe that was connected through a USB port to the conductivity meter, which immediately read the data (Figure 1). 

Saliva specimens were collected according to the following steps (Figure 2):The participant swallowed to empty their mouth.Saliva was collected with a 1-mL dropper, and the collected salivary sample was placed in a 1.5 mL Eppendorf tube.The saliva sample (100 μL) was then diluted with 100 μL of deionized water.Saliva conductivity was then analyzed using the developed portable monitor with a disposable printed-circuit board (PCB) electrode.Data on salivary conductivity was then recorded.

### 2.5. Blood and Urine Collection and Analysis

Blood samples were collected from a vein in the non-dominant hand. All laboratory values, including blood cell count, biochemical data, and copeptin, were analyzed using automated and standardized methods. Urine samples were collected into universal containers and analyzed with standard laboratory methods using an automatic analyzer (SYSMEX-XN9000, Japan; Beckman Coulter-DXC800, Trinity-Premier HB9210, and Advanced Instruments-Osmometer Model 3250, U.S.A; FUJIREBIO-FR-ACE color and ELISA Reader: TECAN Sunrise, Austria). Serum copeptin was analyzed using the Human Copeptin CPP ELISA Kit (P01185 (UniProt, ExPASy)), Abbexa LLC, Houston, TX, USA.

### 2.6. Evaluation of the Thirst Intensity Scales

Two thirst intensity scales were utilized in this study: the categorical scale and visual analog scale [19,20]. The categorical scale reflects the participants’ thirst intensity through seven options, while the visual analog scale uses a linear scale with a range of 0–100 mm.

### 2.7. Statistical Analysis

Continuous variables are presented as the mean ± standard deviation. For comparison between the two groups, the Wilcoxon signed-rank test was used for quantitative variables. The Friedman test was used to compare data among the baseline, water restriction, and rehydration groups. Receiver operating characteristic (ROC) curve analysis was used to quantify the accuracy of medical diagnostic tests among the different groups. The criterion value was determined using the Youden index [21]. The criterion for rejecting the null hypothesis was the 95% confidence interval. Statistical analysis was performed using the Statistical Program for the Social Sciences version 25 (IBM Corporation, Armonk, NY, USA) for Macintosh.

## 3. Results

### 3.1. Demographics of the Participants

This study included 20 voluntary participants from the Chiayi Chang Gung Memorial Hospital (Table 1). The participants were aged 35.15 ± 4.09 years. The males and females had a body height of 175.03 ± 4.06 cm and 162.33 ± 8.25 cm (*p* = 0.000), respectively, and a body weight of 76.84 ± 10.80 kg and 61.89 ± 14.62 kg (*p* = 0.011), respectively. There was no statistically significant difference between the male and females in the body mass index (25.07 ± 3.30 vs. 23.37 ± 3.99, respectively; *p* = 0.123). Laboratory data of estimated glomerular filtration rate and glycohemoglobin levels were within the normal ranges in all participants. 

### 3.2. Biochemical, Urinary, and Hematological Data 

Table 2 shows the biochemical parameters, urinary analysis, and hematological parameters in the three different study phases. The urine creatinine and urine sodium levels increased after water restriction and decreased after rehydration with 1000 mL of water. The concentration of red blood cell count and hemoglobin showed the same trend as the results of the urinary analysis.

### 3.3. Trends of the Different Biomarkers and Parameters in Changing Fluid Status

Compared with the group with a normal status (Figure 3), laboratory data in the group with a 12-h water restriction revealed increased salivary conductivity (3671.445 ± 1161.93 μs/cm vs. 3263.141 ± 1140.30 μs/cm, *p* = 0.018), serum osmolality (293.25 ± 4.30 mOsm/kgH_2_O vs. 290.35 ± 4.08 mOsm/kgH_2_O, *p* = 0.007), urine osmolality (787.80 ± 200.00 mOsm/kgH_2_O vs. 451.95 ± 243.96 mOsm/kgH_2_O, *p* = 0.001), urine specific gravity (1.02 ± 0.01 versus 1.01 ± 0.01, *p* = 0.000), fraction excretion of sodium (0.38 ± 0.20% vs. 0.92 ± 0.45%, *p* = 0.001) and serum copeptin level (254.31 ± 100.05 pg/mL vs. 221.56 ± 89.87 pg/mL, *p* = 0.001). After rehydration with 1000 mL of water, all the levels of the biomarkers mentioned above decreased significantly (Table 3). The same trend was noted in the levels of angiotensin-converting enzyme and aldosterone, but the difference was not statistically significant. The results of the categorical and visual analog scales for thirst intensity significantly differed among the three groups. The serum copeptin levels were not significantly different between the water restriction and rehydration groups. The same clinical trend in different parameters for evaluation of fluid status was observed in different sexes, respectively (Appendix A).

### 3.4. Use of Salivary Conductivity as a Screening Biomarker in Patients with Body Fluid Deprivement

The ROC curve analysis of the salivary conductivity data of the 20 participants (AUC = 0.707, 95% confidence interval 0.542–0.873, *p* = 0.025) (Figure 4) revealed that salivary conductivity had a good sensitivity in distinguishing between the body fluid status during water restriction and rehydration. The optimal salivary conductivity criterion value was 2678.09 μs/cm, which had a sensitivity and specificity of 90% and 55%, respectively.

## 4. Discussion

The analytical data from this study suggest that salivary conductivity was significantly correlated with fluid status in healthy adults. Salivary conductivity reflected the acute changes in fluid status, especially during the rehydration process. This salivary biomarker may be added to the evaluation methods of fluid status. In addition, this study also found that the levels of serum copeptin and the fractional excretion of sodium were sensitive to the acute changes in fluid status. ROC curve analysis provides the criterion values of salivary conductivity for diagnosing dehydration.

According to Lacey et al. [22], there are two main mechanisms for the restoration of fluid balance in adults: decreased total body water and dehydration. First, increased serum osmolality stimulates the osmoreceptor and hypothalamus for the following regulation process. In contrast, decreased blood volume can activate the baroreceptor-mediated sympathetic nervous system and the renin–angiotensin–aldosterone system to maintain fluid status. Both mechanisms are attributed to the increased arginine vasopressin secretion, renal conservation, thirst sensation, and oral intake.

Many biomarkers have been described in the evaluation of fluid status in people with different clinical conditions. In a study of 18 volunteers, Popowski et al. [13] reported that serum osmolality can accurately identify the hydration state and is sensitive to changes in the hydration state during acute dehydration and rehydration. Decreased water intake resulted in increased serum osmolality, which has also been observed in other clinical studies [23]. Urine specific gravity and urine osmolality are also very sensitive to changes in hydration status, but these lag behind during rapid body fluid turnover. As such, they are only moderately correlated with serum osmolality during acute dehydration. The current consensus considers urine osmolality a valid biomarker for diagnosing dehydration [24,25,26]. 

Thirst intensity, assessed using the category and visual analog scales, was used to evaluate the subjective sensation of the fluid status of healthy individuals in daily life [27,28]. According to Davies et al. [29], changes in body weight can be used to estimate body fluid status in ill patients. However, total body weight is not suitable for monitoring the dynamic body fluid status [30,31]. Lemetais et al. [32] revealed that the level of copeptin was associated with fluid status. 

In our study (Figure 2), serum osmolality, urine osmolality, urine specific gravity, thirst intensity category scale and thirst intensity visual analog scale increase after 12-h water restriction and decrease after water rehydration. Fractional excretion of sodium reveals a trend contrasting the other biomarkers above. More serum sodium was preserved by renal regulation to decreased loss of body water during the period of water restriction. The change of biomarkers for evaluation of fluid status in our study is compatible with previous studies. However, the concentration of serum copeptin increases after 12-h water restriction but does not change after water rehydration. It may indicate that serum copeptin does no change immediately during water supplement in the clinical practice.

Furthermore, urine osmolality and urine specific gravity were correlated with copeptin levels. Muñoz et al. [33] reported that salivary osmolality is the most accurate method among serum osmolality, salivary osmolality, urine osmolality, urine volume, and urine specific gravity for detecting dehydration. Several clinical studies have reported that salivary osmolality increased with progressive fluid depletion and dehydration [34,35,36]. In specific clinical conditions, such as hypertonic–hypervolemia dehydration, the sensitivity of salivary osmolality is nearly equal to the osmolality of urine in reflecting fluid status [35]. Currently, salivary osmolality may be a good biomarker for the evaluation of body fluid status [37]. 

In contrast, shorter laboratory turnaround times can prevent patients from being diagnosed [38]. In a prospective study of 166 patients, Moon et al. [39] reported that the median turnaround time (TAT) of routine biochemical tests was 36–57 min, depending on the month. Another prospective study of 81 patients in the emergency department revealed that the median TAT of urine analysis observed via a photometric analyzer was 33 min [40]. Prolonged TAT may lead to delayed diagnosis and consequently, poor clinical outcomes. Therefore, early interventions for dehydration depend on the quick evaluation of the fluid status.

In this study, compared to the rehydration group, both objective and subjective data from the baseline status, including salivary conductivity, serum osmolality, urine osmolality, urine specific gravity, and the thirst intensity scales, tended to reveal the status of water depletion (Figure 3). Symptoms of early dehydration are not clear, and some may be too mild to present with obvious signs [41]. A symptomless water-deprived status should be considered. Neglected mild dehydration may prevent proper fluid intake in daily life.

Responsiveness to changes in fluid intake volume during hydration and rehydration of biomarkers is vital. According to current research, analysis of serum and urine samples can accurately reflect fluid status. However, prolonged TAT may cause poor outcomes; it may take about half to one hour from sample collection to presentation of the valid report [39,40]. Furthermore, an invasive procedure is necessary for blood sample collection. Suboptimal sample collection and transport can lead to incorrect diagnoses in clinical practice [42]. As for the thirst scale, both the categorical and visual analog scales were based on the subjective descriptions of the participants, which may be prone to evaluation bias. Such a subjective evaluation scale is limited by cognitive function and mental status [43].

The use of a portable device for monitoring salivary conduction can provide rapid, real-time, and straightforward laboratory data for the evaluation of fluid status. This specific salivary biomarker can be applied in many fields, such as in farmlands for farmers, pre-operation rooms for short-term fasting patients, sports fields for athletes, medical institutions for patients who have a potential risk of dehydration, and all healthy adults for monitoring fluid status in daily life. Salivary conduction can also provide recommendations to supplement fluid intake.

Finally, the limitations of this study included the small sample size that consisted only of adults. The process of dehydration in this study is a stress test. The clinical condition can be dramatic or unpredictable change during water restriction in children, the elder and patients with systemic disease. Under research ethics, we were prone to selecting the participants who with enough tolerance under the designed stress test in this study. In addition, the short observation period in this study led to the limited information that could have been used for further analysis. Further well-designed prospective studies are warranted to confirm our findings.

## 5. Conclusions

In summary, salivary conductivity is a rapid, real-time, and easy-to-administer screening tool for the evaluation of fluid status in healthy adults. It can be applied to our daily lives to prevent unexpected imbalances in body fluids. This portable monitoring system allows for a rapid and non-invasive method for assessing hydration status, reducing unnecessary medical expenses and preventing the complications of dehydration.

## Figures and Tables

**Figure 1 jpm-11-00577-f001:**
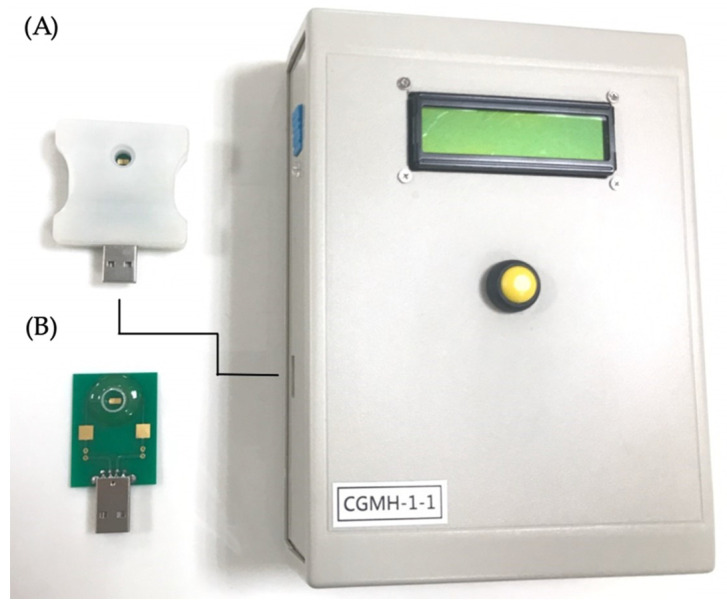
Measuring of salivary conductivity with the portable biodevice. (**A**) Salivary samples were applied to the well of the sensing probe. The sensing probe was connected to the conductivity meter through a USB port and read out the data immediately after pressing the yellow button. (**B**) The disposable printed-circuit board (PCB )of the sensing probe inside the shelter and the electrode were placed in horizontally parallel.

**Figure 2 jpm-11-00577-f002:**
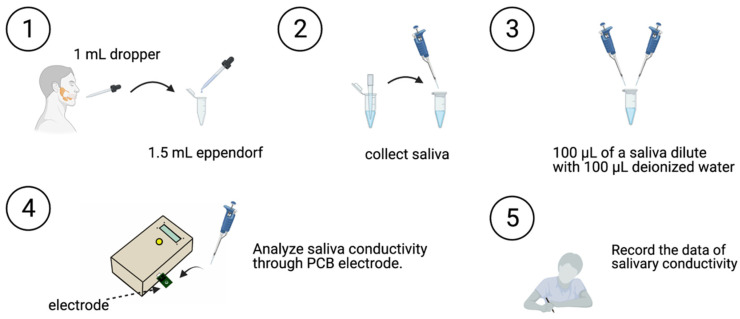
The salivary collection and analysis protocol. Salivary conductivity is detected using a printed-circuit board biodevice. ①: Saliva was collected with a 1-mL dropper. ②: The collected salivary sample was placed in a 1.5 mL Eppendorf tube. ③: The saliva sample (100 μL) was then diluted with 100 μL of deionized water. ④: Saliva conductivity was then analyzed using the developed portable monitor with a disposable printed-circuit board (PCB) electrode. ⑤: Data on salivary conductivity was then recorded.

**Figure 3 jpm-11-00577-f003:**
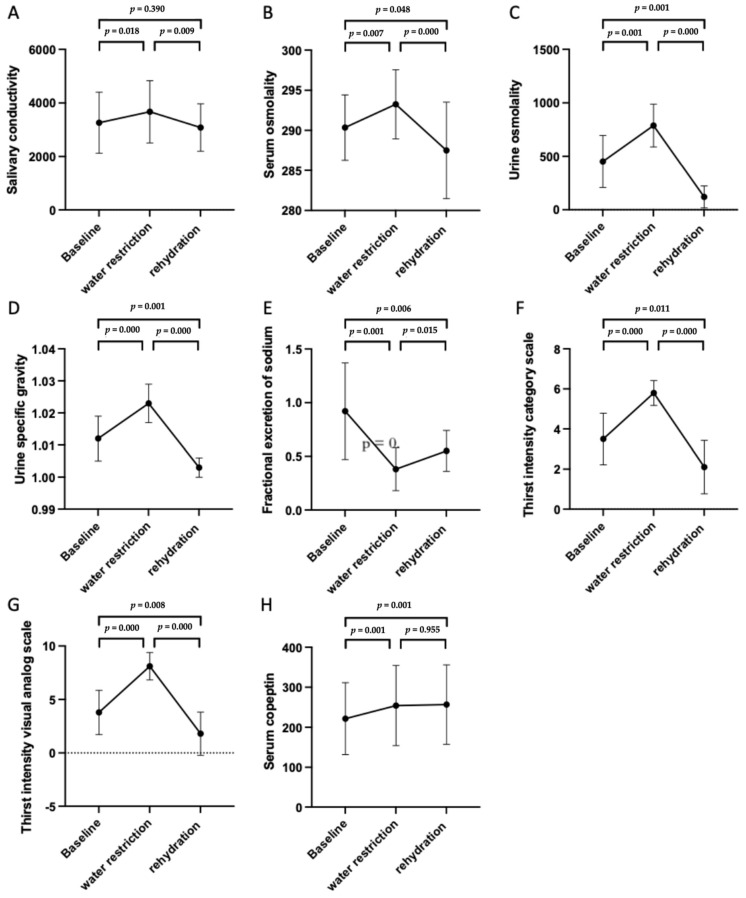
Changes in different biomarkers and parameters for evaluation of fluid status. The changes in the different biomarkers are shown with polygonal line graphs (n = 20), including salivary conductivity (**A**), serum osmolality (**B**), urine osmolality (**C**), urine specific gravity (**D**), fractional excretion of sodium (**E**), thirst intensity category scale (**F**), thirst intensity visual analog scale (**G**), and serum copeptin (**H**). Note: The black dots and error bars indicate the mean and standard deviation of the biomarkers. The *p*-values between two different fluid statuses are labeled in each figure.

**Figure 4 jpm-11-00577-f004:**
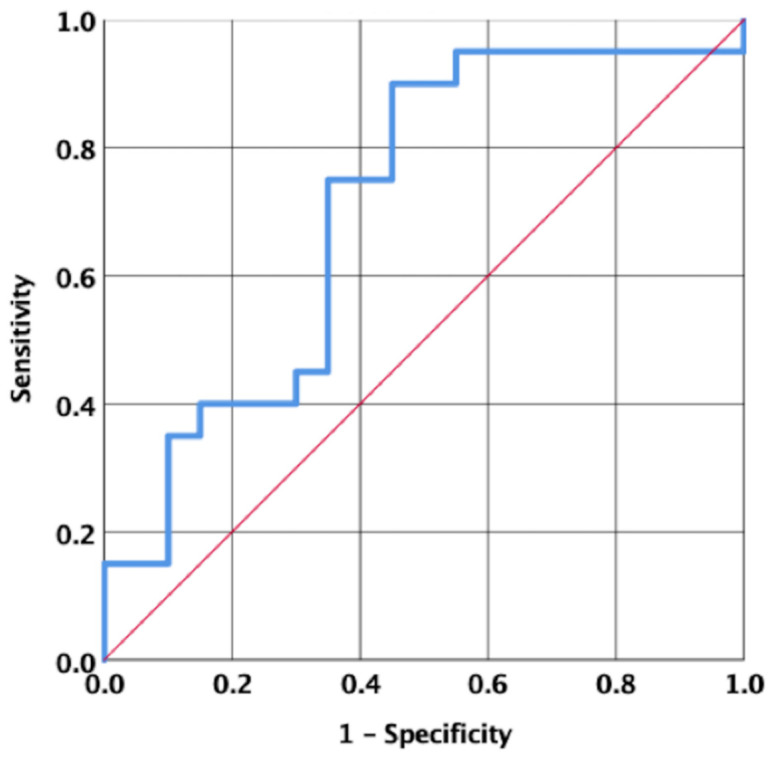
The receiver operating characteristic curve analysis. The receiver operating characteristic curve analysis of salivary conductivity is used to distinguish between different body fluid statuses (water restriction and rehydration statuses); AUC = 0.707, *p* = 0.025.

**Table 1 jpm-11-00577-t001:** Baseline characteristics of the healthy participants, stratified by sex (*n* = 20).

Variables	All patients (N = 20)	Male (N = 10)	Female (N = 10)	*p* Value
Age, years	35.15 ± 4.09	35.50 ± 4.48	34.80 ± 3.88	0.684
HbA1c, %	5.43 ± 0.26	5.43 ± 0.29	5.43 ± 0.25	0.280
eGFR, mL/min/1.73 m^2^	96.12 ± 16.67	91.70 ± 16.90	100.54 ± 16.04	0.912
Height, cm	168.68 ± 9.08	175.03 ± 4.06	162.33 ± 8.25	0.000 ***
Weight, kg	69.37 ± 14.67	76.84 ± 10.80	61.89 ± 14.62	0.011 *
BMI, kg/m^2^	24.22 ± 3.67	25.07 ± 3.30	23.37 ± 3.99	0.123

Note: Values are shown as the mean ± standard deviation. Abbreviations: BMI: body mass index. eGFR: estimated glomerular filtration rate. HbA1c: glycosylated hemoglobin. Note: * *p* < 0.05, *** *p* < 0.001

**Table 2 jpm-11-00577-t002:** Laboratory findings in the different groups (normal status, water restriction, and rehydration status).

Variables	Baseline (Day 1, 0600 p.m.)	12-h Water Restriction (Day 1, 06:00 p.m. to Day 2, 06:00 a.m.)	1-h Water Rehydration (Day 2, 06:00 a.m. to Day 2, 07:00 a.m.)	*p* Value
Biochemical parameters				
Blood urea nitrogen, mg/dL	11.63 ± 1.95	11.17 ± 2.11	10.89 ± 1.82	0.279
Creatinine, mg/dL	0.8335 ± 0.18	0.8240 ± 0.16	0.7835 ± 0.17	0.001 ***
Sodium, mEq/L	141.15 ± 2.08	141.95 ± 1.82	140.05 ± 1.99	0.001 ***
eGFR, mL/min/1.73 m^2^	96.12 ± 16.67	96.35 ± 13.32	103.10 ± 17.11	0.001 ***
Serum glucose, mg/dL	93.45 ± 14.04	84.10 ± 8.84	85.80 ± 10.27	0.010 **
HbA1c, %	5.43 ± 0.26	5.42 ± 0.26	5.42 ± 0.24	0.815
eAG, mg/dL	109.2 ± 7.46	108.70 ± 7.51	108.90 ± 6.77	0.815
Urinary analysis				
Urine creatinine, mg/dL	87.59 ± 81.39	206.80 ± 96.26	22.96 ± 21.68	0.000 ***
Urine sodium, mEq/L	94.9 ± 50.08	117.55 ± 46.98	28.55 ± 21.05	0.001 ***
Hematological parameters				
RBC count, 1000/uL	4.92 ± 0.66	5.01 ± 0.64	4.94 ± 0.74	0.014 *
Hemoglobin, g/dL	13.90 ± 1.40	14.18 ± 1.51	13.96 ± 1.66	0.018 *
Hematocrit, %	41.19 ± 3.57	42.00 ± 4.00	41.23 ± 4.26	0.002 **

Note: Values are shown as the mean ± standard deviation (SD). Abbreviations: eAG: estimated average glucose. eGFR: estimated glomerular filtration rate. HbA1c: glycosylated hemoglobin. RBC count: red blood cell count. Note: * *p* < 0.05, ** *p* < 0.01, *** *p* < 0.001.

**Table 3 jpm-11-00577-t003:** Different parameters for the evaluation of body fluid status.

Variables	Baseline (Day 1, 06:00 p.m.)	12-h Water Restriction (Day 1, 06:00 p.m. to Day 2, 06:00 a.m.)	1-h Water Rehydration (Day 2, 0600 a.m. to Day 2, 07:00 a.m.)	*p* Value
Salivary conductivity, μs/cm	3263.141 ± 1140.30	3671.445 ± 1161.93	3081.006 ± 884.62	0.019 *
Serum osmolality, mOsm/kgH_2_O	290.35 ± 4.08	293.25 ± 4.30	287.50 ± 6.02	0.000 ***
Urine Osmolality, mOsm/kgH_2_O	451.95 ± 243.96	787.80 ± 200.00	119.20 ± 102.25	0.000 ***
Urine SG	1.012 ± 0.007	1.023 ± 0.006	1.003 ± 0.003	0.000 ***
Thirst intensity CS	3.50 ± 1.28	5.80 ± 0.62	2.10 ± 1.33	0.000 ***
Thirst intensity VAS	3.79 ± 2.07	8.11 ± 1.28	1.80 ± 2.02	0.000 ***
Serum copeptin, pg/mL	221.56 ± 89.87	254.31 ± 100.05	256.74 ± 99.24	0.001 **
FeNa, %	0.92 ± 0.45	0.38 ± 0.20	0.55 ± 0.19	0.000 ***
Weight, kg	69.37 ± 14.67	76.84 ± 10.80	61.89 ± 14.62	0.011 *
ACE test, IU/L	15.35 ± 4.10	15.81 ± 4.01	15.48 ± 3.91	0.779
Aldosterone, ng/dL	10.877 ± 8.17	11.37 ± 6.77	10.32 ± 5.87	0.861
Renin, ng/L	18.196 ± 13.79	16.25 ± 8.76	14.75 ± 7.03	0.287

Note: Values are shown as the mean ± standard deviation (SD). Abbreviations: ACE test: angiotensin-converting enzyme test. FeNa: fractional excretion of sodium. Thirst intensity CS: thirst intensity categorical scale. Thirst intensity VAS: thirst intensity visual analog scale. Urine SG: urine specific gravity. Note: * *p* < 0.05, ** *p* < 0.01, *** *p* < 0.001.

## Data Availability

The datasets used and analyzed in this study are available from the corresponding author upon reasonable request.

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
