# Peer review of "A Portable Biodevice to Monitor Salivary Conductivity for the Rapid Assessment of Fluid Status"

_jpm, 2021, doi:10.3390/jpm11060577_

Round 1
Reviewer 1 Report
The authors detected the salivary conductivity and serum and urine biomarkers of 20 volunteers using the portable monitoring system and laboratory experiments. They claimed that salivary conductivity was significantly correlated with fluid status in healthy adults.
Some comments:
- There are significant differences in height and weight between males and females among the 20 volunteers. The authors should compare the effect of gender on the results of the study.
- There are some confusing points in Tables 2 and 3. For example, what does this mean? “1800 h, 0600 h, 0700 h.”
- In the discussion, the authors don’t sufficiently discuss the results of this paper, but present too many research advances in the field.
- In the conclusion, the authors mentioned “portable monitoring system allows for a reliable and non-invasive method for assessing hydration status”. However, it is not the conclusion of this manuscript, but more like the conclusions of other study (ref.15, Lu, Y.P.; Huang, J.W.; Lee, I.N.; Weng, R.C.; Lin, M.Y.; Yang, J.T.; Lin, C.T. A portable system to monitor saliva conductivity for dehydration diagnosis and kidney healthcare. Sci Rep 2019, 9, 14771, doi:10.1038/s41598-019-51463-8.).
Author Response
Response to Reviewer 1 Comments
Point 1: There are significant differences in height and weight between males and females among the 20 volunteers. The authors should compare the effect of gender on the results of the study.
Response 1: Thank you for your valuable advice. We had compared the effect of sex on the results of this study. There were no significant differences for each biomarker in this study between males and females except for the level of serum osmolality in normal status (P=0.007) (Table S1, Table S2, Table S3). The same clinical trend of change in different parameters was observed in different sex respectively. The following tables would be placed as supplemental material. We had revised our manuscript for the effect of sex in the part of result. (page 6, lines 208-210)
Table S1. Different parameters for the evaluation of body fluid status in normal status.
Variables |
Male (N=10) |
Female (N=10) |
P value |
Salivary conductivity, μs/cm |
3263.49±467.62 |
3262.79±1589.46 |
0.096 |
Serum osmolality, mOsm/kgH2O |
292.60±2.59 |
288.10±4.15 |
0.007** |
Urine Osmolality, mOsm/kgH2O |
544.90 ±250.47 |
359.00 ±209.07 |
0.070 |
Urine SG |
1.015 ±0.007 |
1.009 ±0.006 |
0.069 |
Thirst intensity CS |
3.60±1.17 |
3.40±1.42 |
0.785 |
Thirst intensity VAS |
3.75±1.89 |
3.82±2.34 |
0.821 |
Serum copeptin, pg/mL |
198.89±78.08 |
244.25±99.03 |
0.496 |
FeNa, % |
0.93±0.48 |
0.91±0.44 |
0.940 |
Note: Values are shown as the mean± standard deviation (SD). Abbreviations: FeNa: fractional excretion of sodium. Thirst intensity CS: thirst intensity categorical scale. Thirst intensity VAS: thirst intensity visual analog scale. Urine SG: urine specific gravity. Note: *P < 0.05, **P < 0.01, ***P<0.001.
Table S2. Different parameters for the evaluation of body fluid status in water restriction.
Variables |
Male (N=10) |
Female (N=10) |
P value |
Salivary conductivity, μs/cm |
3867.96±1133.77 |
3474.93±1216.11 |
0.364 |
Serum osmolality, mOsm/kgH2O |
294.80±2.74 |
291.70±5.12 |
0.129 |
Urine Osmolality, mOsm/kgH2O |
831.10 ±150.86 |
744.50 ±239.83 |
0.450 |
Urine SG |
1.024 ±0.004 |
1.021 ±0.006 |
0.239 |
Thirst intensity CS |
5.60±0.70 |
6.00±0.47 |
0.118 |
Thirst intensity VAS |
7.85±1.62 |
8.37±0.83 |
0.570 |
Serum copeptin, pg/mL |
219.46±81.69 |
289.17±108.43 |
0.112 |
FeNa, % |
0.37±0.18 |
0.40±0.23 |
0.940 |
Note: Values are shown as the mean± standard deviation (SD). Abbreviations: FeNa: fractional excretion of sodium. Thirst intensity CS: thirst intensity categorical scale. Thirst intensity VAS: thirst intensity visual analog scale. Urine SG: urine specific gravity. Note: *P < 0.05, **P < 0.01, ***P<0.001.
Table S3. Different parameters for the evaluation of body fluid status in rehydration status.
Variables |
Male (N=10) |
Female (N=10) |
P value |
Salivary conductivity, μs/cm |
3201.82±897.66 |
2960.20±902.13 |
0.140 |
Serum osmolality, mOsm/kgH2O |
289.80±1.69 |
285.20±7.87 |
0.139 |
Urine Osmolality, mOsm/kgH2O |
110.40 ±69.04 |
128.00 ±130.89 |
0.880 |
Urine SG |
1.003 ±0.002 |
1.004 ±0.004 |
0.691 |
Thirst intensity CS |
2.30±1.64 |
1.90±0.99 |
0.749 |
Thirst intensity VAS |
2.53±2.63 |
1.56±1.38 |
0.596 |
Serum copeptin, pg/mL |
239.00±98.74 |
274.48±101.70 |
0.623 |
FeNa, % |
0.63±0.23 |
0.48±0.10 |
0.112 |
Note: Values are shown as the mean± standard deviation (SD). Abbreviations: FeNa: fractional excretion of sodium. Thirst intensity CS: thirst intensity categorical scale. Thirst intensity VAS: thirst intensity visual analog scale. Urine SG: urine specific gravity. Note: *P < 0.05, **P < 0.01, ***P<0.001.
Point 2: There are some confusing points in Tables 2 and 3. For example, what does this mean? “1800 h, 0600 h, 0700 h.”
Response 2: Thank you for the question. “1800 h, 0600 h, 0700 h.” is indeed hard to understand the presentation of time. It means the point of time that we conduct our study. 1800 h means 06:00 p.m.; 0600 h means 06:00 a.m.; 0700 h means 07:00 a.m.. We had revised our manuscript with standard presentation of time. (page 5, lines 191; page 7, lines 217)
Point 3: In the discussion, the authors don’t sufficiently discuss the results of this paper, but present too many research advances in the field.
Response 3: Thank you for your recommendation. We strength the content of the discussion. Manuscript had been revised based on your recommendation. (page 9, lines 264-273)
Point 4: In the conclusion, the authors mentioned “portable monitoring system allows for a rapid and non-invasive method for assessing hydration status”. However, it is not the conclusion of this manuscript, but more like the conclusions of other study (ref.15, Lu, Y.P.; Huang, J.W.; Lee, I.N.; Weng, R.C.; Lin, M.Y.; Yang, J.T.; Lin, C.T. A portable system to monitor saliva conductivity for dehydration diagnosis and kidney healthcare. Sci Rep 2019, 9, 14771, doi:10.1038/s41598-019-51463-8.).
Response 4: Thank you for pointing this out. Indeed, this study share similar concept with the previous study (Sci Rep 2019, 9, 14771). In this study, the portable monitoring system showed good sensitivity in distinguishing between the body fluid status after water restriction and rehydration. Collection of saliva was through a non-invasive method. Besides, in our study, salivary conductivity can reflect continuous and real-time change of fluid status in participants. Therefore, “portable monitoring system allows for a rapid and non-invasive method for assessing hydration status” is also an important concept of this study. In previous study of 30 participants, Lu et al selected 10 healthy adults, 10 healthy farmers and 10 patients with chronic kidney disease for clinical research at the same period. However, this study selected 20 healthy adults for continuous monitoring. We aimed to clarify that salivary can be a valid biomarker for evaluation of fluid status for individuals. Furthermore, we decided to revise our conclusion form “diagnostic tool” to “screening tool” and from “reliable and non-invasive method” to “rapid and non-invasive method”. We had revised our manuscript according to the previous discussion. (page 10, lines 324-325; page 10, lines 327)
Please see the attachment. Thank you.

Reviewer 2 Report
Thank you for giving me to review your manuscript. This manuscript is interesting and scientifically meaningful for considering the volume status of patients with dehydration. However, there are various flaws. I suggest several revisions.
- In the abstract, this device's specificity regarding detection of dehydration is very low (specificity: 55%), so the diagnosis of dehydration by this device is not helpful. This study should collect more participants for checking the validity of this test or change the cutoff of the test for the better specificity.
- In the introduction, there is no clear theoretical framework and the comparison of other tests such as medical history and physical examinations. Based on the previous studies, there are history and physical examinations with high sensitivity and specificity. This article should include the descriptions regarding such findings.
- In the sample section of the method, there are no descriptions regarding sample calculation. The authors should descript why they chose 20 as the sample size.
- In the method section's measurement, what do the authors mean about systemic diseases? This study should clearly describe inclusion and exclusion criteria.
- In the method section, the authors should describe what the participants did during 12 hours. The condition of water in bodies can depend on their activities. Their activities and limitations should be described.
- In the analysis section, the validation study regarding devices should include more participants regarding various backgrounds. This study has few participants with less diversity.
- The discussion part should be revised based on the previous revision.
Author Response
Response to Reviewer 2 Comments
Point 1: In the abstract, this device's specificity regarding detection of dehydration is very low (specificity: 55%), so the diagnosis of dehydration by this device is not helpful. This study should collect more participants for checking the validity of this test or change the cutoff of the test for the better specificity.
Response 1: Thank you for pointing this important concept out. This portable biodevice to monitor salivary conductivity for assessment of fluid status is aim to be designed as a rapid and non-invasive screening tool. As a screening tool, high sensitivity is more important for clinical practice. With high sensitivity (90%) of this biodevice, we can find out the patient who is suffered from dehydration in clinical practice. Besides, we had revised our conclusion in the manuscript from “diagnostic tool” to “screening tool” and from “reliable and non-invasive method” to “rapid and non-invasive method”. (page 10, lines 324-325; page 10, lines 327)
Point 2: In the introduction, there is no clear theoretical framework and the comparison of other tests such as medical history and physical examinations. Based on the previous studies, there are history and physical examinations with high sensitivity and specificity. This article should include the descriptions regarding such findings.
Response 2: Thank you for your valuable advice. Medical history and physical examination have been used for evaluation and diagnosis of dehydration. Medical history and physical examination are useful for evaluation in specific clinical condition, such as gastroenteritis in children, but not suitable for applying to general clinical practice. Medical history can provide information of possible water depletion but cannot provide quantitative data objectively. Besides, previous study revealed that physical examination had poor sensitivity (0-44%). There no valid index for evaluation of dehydration via observing physical signs of dehydration [tachycardia (>100 bpm), low systolic blood pressure (<100 mm Hg), dry mucous membrane, dry axilla, poor skin turgor, sunken eyes, and long capillary refill time (>2 seconds) (J Am Med Dir Assoc. 2015 Mar;16(3):221-8.). Other study also indicated that there is limited evidence of the diagnostic utility of individual clinical symptom and sign. (Am J Clin Nutr. 2016 Jul;104(1):121-31.). A prospective diagnostic accuracy study reported that commonly used signs and symptoms of dehydration lack even basic levels of diagnostic accuracy. We had revised our manuscript according to your recommendation for supplement of our introduction. (J Am Med Dir Assoc. 2019 Aug;20(8):963-970.). (page 2, lines 65-70)
Point 3: In the sample section of the method, there are no descriptions regarding sample calculation. The authors should descript why they chose 20 as the sample size.
Response 3: Thank you for your recommendation. We used the significance level of 5%, the 80% power of this study and a hypothetical effect size of 0.6 (large effect). The required sample would be 20 participants. Post-hoc computing of the power of study and effect size of all the outcomes was performed using the G-power statistical software. We had revised the part of method in the manuscript. (page 3, lines107-109)
Point 4: In the method section's measurement, what do the authors mean about systemic diseases? This study should clearly describe inclusion and exclusion criteria.
Response 4: Thank you for pointing this out. Systemic disease concerned in this study including any cardiovascular disease, peripheral vascular disease, renal disease, hepatic disease, infection episode and endocrine disease. Manuscript had been revised according to your comment. (page 3, lines 104-106)
Point 5: In the method section, the authors should describe what the participants did during 12 hours. The condition of water in bodies can depend on their activities. Their activities and limitations should be described.
Response 5: We thank the reviewer for this comment. The participants were prohibited to drink any fluid nor eat food containing more than 500cc fluid for 12 hours to achieve a dehydration status. There was no limitation of their physical activities and the status of dehydration was confirmed by serum and urinary analyses after 12 hours. We had revised our experimental procedure in the manuscript as description above. (page 3, lines 115-119)
Point 6: In the analysis section, the validation study regarding devices should include more participants regarding various backgrounds. This study has few participants with less diversity.
Response 6: Thank you for this recommendation. The process of dehydration in this study is a stress test. The clinical condition can be dramatic or unpredictable change during water restriction process in children, the elder and patients with systemic disease. Under research ethics, we prone to enroll the participants with enough tolerance under the designed stress test in this study. That is the reason why we chose healthy adults for this pilot study. We had revised our manuscript at discussion part as limitation of this study (page 10, lines 316-319)
Point 7: The discussion part should be revised based on the previous revision.
Response 7: We had revised our manuscript based on our response of each comment.
Please see the attachment. Thank you.

Reviewer 3 Report
In general, this manuscript is written well, and the data support their conclusion, so it can be published after some minor issues are addressed. These issues are listed below.
- I hope the authors can introduce more details on this portable device, assumably shown in Figure 1 with the PCB electrode. It will be great if they can use additional figure to explain the structure of the PCB electrode and how this device works.
- Figure 2 seems to have lots of information, may deserve more discussions. For example, the standard deviation and p-values are very different for each biomarker, the changing trend are also different. Please add more explanations to each plot.
- The sample size is small, so is there any better statistical method can be used to analyze the data so that we can have a clearer idea of the sensitivity and selectivity? Maybe Bayesian method can help? If this paper just focuses on the detection process, not the biomedical mechanism, probably it is ok. That is why the authors can emphasize more on the device part.
Author Response
Please see the attachement. Thank you.

Round 2
Reviewer 1 Report
no comments
Reviewer 2 Report
The manuscript has been considerably improved. I think that this paper is suited for inclusion in our journal.
This manuscript is a resubmission of an earlier submission. The following is a list of the peer review reports and author responses from that submission.